# Self-directed digital interventions for the improvement of emotion regulation—effectiveness for mental health and functioning in adolescents: protocol for a systematic review

Abigail Thomson [1], Erin G Lawrence,[1] Bonamy R Oliver,[2] Ben Wright,[3,4] Georgina M Hosang[1]

¹Centre for Psychiatry and Mental Health, Wolfson Institute of Population Health, Queen Mary University of London, London, UK
²UCL Institute of Education, University College London, London, UK
³East London NHS Foundation Trust, London, UK
⁴City University of London, London, UK

**Correspondence to**
Abigail Thomson;
a.c.thomson@qmul.ac.uk

## ABSTRACT

**Introduction** Research suggests that problems with emotion regulation, that is, how a person manages and responds to an emotional experience, are related to a range of psychological disorders (eg, bipolar disorder, anxiety and depression). Interventions targeting emotion regulation have been shown to improve mental health in adults, but evidence on related interventions for adolescents is still emerging. Increasingly, self-directed digital interventions (eg, mobile apps) are being developed to target emotion regulation in this population, but questions remain about their effectiveness. This systematic review aimed to synthesise evidence on current self-directed digital interventions available to adolescents (aged 11–18 years) and their effectiveness in addressing emotion regulation, psychopathology and functioning (eg, academic achievement).

**Methods and analysis** Several electronic databases will be searched (eg, MEDLINE, PsycINFO, ACM Digital Library) to identify all studies published any time after January 2010 examining self-directed digital interventions for adolescents, which include an emotion regulation component. This search will be updated periodically to identify any new relevant research from the selected databases. Data on the study characteristics (eg, author(s)) and methodology, participant characteristics (eg, age) and the digital interventions used to address emotion (dys-) regulation (eg, name, focus) will be extracted. A narrative synthesis of all studies will be presented. If feasible, the effectiveness data will be synthesised using appropriate statistical techniques. The methodological quality of the included studies will be assessed with the Effective Public Health Practice Project quality assessment tool.

**Ethics and dissemination** Ethical approval is not required for this study. Findings will be disseminated widely via peer-reviewed publications and presentations at conferences related to this field.

**Registration details** PROSPERO CRD42022385547.

## INTRODUCTION

Approximately 75% of mental health problems begin in adolescence.[1] In the UK, one in six adolescents (aged 11–16 years) has been

### STRENGTHS AND LIMITATIONS OF THIS STUDY

⇒ This Preferred Reporting Items for Systematic Review and Meta-Analysis-guided review applies a systematic methodology that includes a comprehensive search strategy and well-defined eligibility criteria for screening.

⇒ We follow a dual-review process; a second independent researcher is employed throughout the screening and data extraction process.

⇒ This review uses the Effective Public Health Practice Project quality assessment tool—a tool validated for reviews of intervention effectiveness—to assess the methodological quality of included studies and comment on the strength of the existing evidence.

⇒ The heterogeneity of the interventions and tools used to measure emotion regulation may not allow for direct comparisons between studies as part of a planned meta-analysis.

identified as having a probable mental health problem—a figure that has steadily increased since 2004.[2] Current attempts to support this population are designed to target specific conditions (eg, depression). However, 60% of those adolescents with one diagnosable mental health problem have one or more additional conditions.[3]

Mental health comorbidity—the presence of two or more mental illnesses in an individual—is the rule, rather than the exception, and has been associated with greater clinical severity and a greater detriment to overall quality of life.[4] There is also significant overlap in the aetiological origins of different psychopathologies, including evidence of major overlap in both genetic and environmental contributors between most disorders (eg, major depressive disorder, bipolar disorder and schizophrenia).[5] Further complexity emerges from evidence linking

non-specific risk factors (such as adverse childhood experiences: ACEs) with the onset of psychopathology.[6] ACEs (eg, abuse and domestic violence) are a robust predictor of various mental and physical disorders and symptoms.[7 8] While the mechanisms underpinning the aetiology and development of different psychopathologies are complex, associations have been found across a range of different conditions, including mood disorders, anxiety disorders and substance use disorders.[6]

## A transdiagnostic approach

Given this, researchers have highlighted the need to understand and develop interventions that are directly effective across several disorders, or which alter psychopathological processes common to multiple conditions; these transdiagnostic strategies would offer broader, more effective support for adolescents struggling with their mental health.[9] The Unified Protocol (UP) for the transdiagnostic treatment of emotional disorders is one such strategy that targets the emotional processes (eg, emotional reactivity, emotional responding) underlying multiple different disorders. The UP is a transdiagnostic, emotion-focused cognitive-behavioural therapy (CBT) designed to treat all anxiety and unipolar mood disorders, as well as other disorders with strong emotional components (eg, dissociative and somatoform disorders[10]). Preliminary evidence has found it an effective treatment for adolescents with emotional disorders,[11 12] demonstrating the possible utility of transdiagnostic approaches in adolescent mental health.

Emerging evidence suggests the transdiagnostic approach could be effective in targeting a range of psychopathologies in young people and can activate a range of related, beneficial developmental cascades, including improvement in social and academic outcomes in adolescence, as well as improved sleeping patterns.[13] A transdiagnostic approach to treatment is also thought to be time-effective and cost-effective compared with disorder-specific strategies and may offer a more sustainable alternative to treatments currently available to this population.[4]

## Emotion regulation—a transdiagnostic target?

Emotion regulation has received increased attention in recent years as a transdiagnostic mechanism and clinical target in psychological treatment.[14] Emotion regulation is a multidimensional process wherein an individual monitors, evaluates and shapes their emotions, when they have them, and how they experience or express them. This process is typically understood to be goal-directed, helping individuals to meet the demands of their environment and achieve their ambitions (eg, remaining calm to resolve a conflict).[15] Gross provides a well-defined 'Process model of emotion regulation'[16 17] of specific relevance for research and practice, whereby an individual recognises an emotion regulation goal, selects and, finally, implements specific emotion regulation strategies.[18] Gross defined a set of five distinct emotion

regulation strategies occurring at different points in an emotional experience: situation selection, situation modification, attentional deployment, cognitive change and response modulation.[18] Situation selection can be understood as an individual's efforts to alter the likelihood of being in an emotion-evoking situation. Situation modification involves modifying a situation at the time to change its emotional impact. Attentional deployment involves directing attention towards or away from an emotion or its causes. Cognitive change enables reappraising a situation to change its emotional significance. Response modulation includes any efforts to modify the behavioural, experiential and physiological elements of an emotional response. Each of these strategies can be understood to influence an individual's emotional response in a way that can be interpreted as adaptive (eg, problem-solving, acceptance) or maladaptive (eg, withdrawal, suppression),[17] depending on the context.

Maladaptive patterns of emotional experience or expression that interfere with goal-directed activity are typically understood as emotion *dysregulation* and have physiological, cognitive and social consequences.[17 18] Emotion dysregulation can also be understood to represent problematic emotion dynamics: persistence, lability and intensity of emotions.[19] Evidence demonstrates emotion dysregulation is present across a range of psychopathologies, including internalising (eg, generalised anxiety disorder, major depressive disorder, dysthymia) and externalising disorders (eg, attention-deficit/hyperactivity disorder, conduct disorder, oppositional defiant disorder).[20] For example, generalised anxiety disorder has been associated with a lack of understanding of emotions and an increased reliance on emotion regulation strategies that could be understood as maladaptive, such as withdrawal.[20] Similarly, attention-deficit/hyperactivity disorder is characterised by individuals' emotion regulation deficits and emotion reactivity.[21]

Most evidence on the impact of emotion regulation on psychopathology has been derived from adult populations.[22] However, recent findings are beginning to show a close association between emotion regulation and psychopathology in adolescence.[23] Evidence indicates a significant shift in emotion regulation between the ages of 13 and 15 (eg, access to strategies, use of adaptive vs maladaptive strategies), suggesting that adolescence is a particularly vulnerable period in the development of emotion regulation.[24] Therefore, interventions targeting emotion regulation as a transdiagnostic construct central to developing and maintaining psychopathology may reduce the risk and severity of adolescent mental health problems.

## Emotion regulation interventions

Existing psychological interventions adopt different approaches to improving emotional regulation. Some focus on reducing the use of maladaptive strategies such as rumination (eg, rumination-focused cognitive behavioural therapy[25]), while others focus on increasing

the use of adaptive strategies such as acceptance (eg, Acceptance and Commitment Therapy[26]). Both the adult and adolescent literature indicate that such interventions improve emotion regulation and mediate decreases in psychopathological symptoms within certain diagnoses (eg, anxiety).[23 27] However, the effectiveness of these interventions in improving multiple mental health and functional outcomes, rather than specific and individual symptom groups, is largely unknown.[28]

Further, much of the research to date has focused on the delivery of such interventions in person despite a growing number of digital solutions for adolescent emotion regulation and psychopathology.[23] Previous research examining the effectiveness of several different digital interventions targeting emotion regulation in adolescents found that, in general, such interventions (eg, digital games and biofeedback) can be effective in improving emotion regulation.[29] Digital interventions also have a greater capacity for innovation and engagement with adolescents[30] and the potential to extend effective care cost-effectively and sustainably,[31] but less is known about how such interventions can be applied at scale to support this population.

### A self-directed, digital approach
An increasing number of innovative *self-directed* digital interventions are being developed, which target emotion regulation and related processes (eg, emotional self-awareness).[32 33] Such interventions (eg, mobile apps) are led by the service user, with little to no support from anyone else (eg, therapist, parent/carer), and aim to widen access to support for adolescents. Despite a growing number of self-directed digital interventions made available to adolescents, the effectiveness of interventions delivered in this self-directed and digital format for improving emotion regulation and psychopathology is, as yet, unclear. Further, more evidence is needed to determine the 'active ingredients' of self-directed digital interventions that improve emotion regulation the most and reduce psychopathology in adolescents.[34] As the prevalence of adolescent mental health problems increases, innovative digital solutions could help to ensure that a broad range of adolescents can access and engage in support for their mental health.

### Review objectives
To address some of these issues, the proposed systematic review will synthesise evidence on current self-directed digital interventions available to adolescents (aged 11–18 years) to investigate their effectiveness in addressing emotion (dys-)regulation, psychopathology and functioning (eg, academic achievement).

Despite an increase in the number of available self-directed digital interventions for emotion regulation, evidence about their effectiveness among adolescent populations has yet to be synthesised. This review provides an important extension to existing work which has thus far demonstrated the effectiveness of in-person or therapist-supported emotion regulation interventions

available for young people (aged 6–24 years),[20] as well as the utility of a broad spectrum of digital emotion regulation interventions for adolescents.[29] This review takes a more specific focus to develop evidence on a burgeoning number of *self-directed*, scalable, digital mental health interventions available to adolescents with or without diagnosed psychopathology. Specifically, we will seek to answer the following research questions:
1. Are current self-directed digital interventions that target emotion regulation effective in improving emotion (dys-)regulation in adolescents?
2. Are these interventions effective in treating psychopathology and improving function (eg, academic achievement)?
3. What are the specific components (eg, mood monitoring) of these interventions that most improve emotion (dys-)regulation and/or psychopathology and functioning in adolescents?

## METHODS AND ANALYSIS
### Protocol and registration
This protocol is written in accordance with the Preferred Reporting Items for Systematic Review and Meta-Analysis Protocols (PRISMA-P) guidelines.[35] The completed PRISMA-P checklist is provided in online supplemental file 1. The systematic review has been registered in PROSPERO, the International Prospective Register of Systematic Reviews (CRD42022385547). Important amendments to this protocol will be reported and published with the review results.

### Eligibility criteria
The following criteria (table 1) are set out to screen identified studies using a Population, Intervention, Comparison, Outcomes and Study (PICOS) framework.[36]

### Information sources
10 electronic databases will be searched for studies published any time from January 2010: MEDLINE, PsycINFO; Global Health: Scopus; Web of Science Core Collection; EBSCO CINAHL; EBSCO ERIC; OVID Embase; The Cochrane Central Register of Controlled Trials (CENTRAL) and ACM Digital Library. Grey literature such as preprints and theses are also included in this review (databases: HMIC, EThOS, Psyarxiv, Trip, ClinicalTrials.gov). This search will be updated periodically to identify any new relevant research from the selected databases before the results are finalised (most recent search in August 2023).

### Search strategy
The search strategy (online supplemental file 1) is designed to identify all studies examining one or more self-directed digital mental health interventions for adolescents, which include an emotion (dys-)regulation component. It will search for synonyms for the following three concepts: *adolescents, emotion (dys-)regulation and self-directed digital interventions.* The search will be limited

**Table 1** Inclusion and exclusion criteria

| | Inclusion | Exclusion |
|---|---|---|
| Population | ▶ Any adolescent (aged 11–18)<br>▶ With or without a diagnosed mental health problem<br>▶ Neurotypical/neurodiverse | ▶ Adolescents outside the specified age range<br>▶ Non-human participants |
| Intervention | ▶ Any mental health intervention delivered entirely by digital technology (eg, app-based, web-based)<br>▶ Can be delivered in any appropriate setting such as at home, in school, in voluntary and community settings, primary, secondary or tertiary care<br>▶ Must address emotion (dys-)regulation, but this need not be the focus<br>▶ Must be self-directed (ie, the intervention is completed entirely by the adolescent, with no support from a clinician, teacher, parent or caregiver) | ▶ The intervention is not entirely self-directed (ie, it includes modules, or components which are not completed by the adolescent, or are completed with support from a clinician, teacher, parent or caregiver) |
| Comparison | ▶ Employing a control group is not an essential criterion for inclusion in the review | ▶ Employing a control group is not an essential criterion for inclusion in the review |
| Outcomes | ▶ Must measure emotion (dys-)regulation using a valid or appropriate measure | ▶ If a valid and appropriate item, scale or measure is not used, the authors of the paper must comment on their reasoning for choosing their measure |
| Study | ▶ Publication dates from 2010 to present<br>▶ Studies from any geographical location<br>▶ Primary research study with direct contact or observation of individuals<br>▶ Grey literature, theses, reports, pre-prints | ▶ Non-English language<br>▶ Descriptive studies such as case reports/studies that do not include direct contact or observation of individuals<br>▶ Review papers, books, editorials, commentaries, poster abstracts, conferences |

to English language studies conducted with a human population.

## Study selection

All identified studies will be exported to Endnote 20, and any duplicates will be removed following a specified method[37] before being imported into Rayyan[38] for further deduplication and screening. Abstracts and titles will be screened based on the inclusion and exclusion criteria (table 1). Those studies that meet these criteria will enter the full-text screening stage for further checks against the eligibility criteria. The full-text screening will allow for the identification of those interventions that do not address emotion (dys-)regulation directly. A second researcher (EGL) will be assigned a random 10% of the identified studies for screening at the title and abstract stage. Discrepancies throughout the study selection process will be resolved through consensus between AT and EGL. If discrepancies cannot be resolved through discussion, a third reviewer will be consulted to adjudicate.

## Data extraction and management

Data will be extracted and collated by two independent reviewers (AT and EGL) and tracked in Microsoft Excel using a structured coding form and associated coding manual. The form will be piloted on a sample of included studies and refined before extraction begins. Information relating to study characteristics (eg, author(s), publication date), participant characteristics (eg, age, gender), digital intervention characteristics (eg, name, focus), and relevant clinical and emotion dysregulation outcomes will be extracted from each study. Information to determine any study bias will also be collated. Study investigators will be contacted for missing/unreported data or additional details. If we are not able to obtain sufficient details/raw data using this approach, we will exclude the respective studies.

## Outcomes

The primary outcome of this review is the change in emotion (dys-)regulation, psychopathology and functioning (eg, academic achievement) occurring as a result of participation in a self-directed digital intervention that addresses emotion (dys-)regulation. Emotion (dys-)regulation must be assessed, and where possible, using a valid and appropriate item, scale or measure (eg, Child Social Behaviour Questionnaire),[39] including through clinical interviews or self-reported measures. An existing review of emotion (dys-)regulation assessment[40] and similar reviews[22 41] will be used as guidance to decide on a measure's eligibility. Symptoms of psychopathology will be assessed by any available valid and appropriate measure, including through clinical interview or self-reported measures.

## Quality and risk of bias assessment

Two researchers (AT and EGL) will independently assess the methodological quality of the included studies using

the Effective Public Health Practice Project Quality Assessment tool (EPHPP). The EPHPP is applicable to a range of quantitative study designs (eg, case-control studies) and has been judged to be particularly suitable for systematic reviews on the effectiveness of interventions/treatments.[42] Evidence has shown the EPHPP has good content and construct validity.[42 43] The AACODS Checklist will be used to assess the quality and content of any included grey literature (ie, theses and reports). It has been designed as a critical appraisal tool specifically for use with grey literature sources.

If sufficient data is available, potential reporting bias will be assessed visually using a funnel plot and commented on in the review. An asymmetrical plot suggests a publication bias.[44] If asymmetry in the funnel plot is detected, the studies will be reviewed to assess whether this asymmetry was likely due to publication bias or other factors such as methodological or clinical heterogeneity.

## Data synthesis

Information from the included studies will be synthesised in line with guidance from Popay.[45] A narrative overview of the study characteristics and methodology, participant characteristics and the digital interventions used to address emotion (dys-)regulation will be included. A summary of any observed changes in emotion dysregulation and/or psychopathology occurring as a result of the intervention will be provided. Additionally, any general improvements in functioning (eg, social or academic) that occur as a result of participation in the intervention will also be described if reported in the included papers.

Given the findings of preliminary searches and previous reviews,[23] it is anticipated that there will be high heterogeneity across studies. However, a meta-analysis will be conducted if (a) there are a minimum of two papers included in the review and (b) the available data within the included studies are sufficiently homogenous (Q statistic is non-significant and/or $I^2 < 25\%$). Studies judged to be of low quality will be excluded from the meta-analysis. Two meta-analyses are planned: one with psychopathology as the primary outcome and one with emotion dysregulation as the primary outcome. Outcomes will be pooled using a random-effects model, in line with the current recommendations for meta-analysis models in psychology.[46]

Subgroup analyses will be completed if sufficient data are available to explore the efficacy of different intervention components (ie, mood monitoring, psychoeducation) in improving emotion dysregulation and psychopathology in adolescents. Subgroup analyses will also be conducted to identify whether there are differences in effect size or heterogeneity due to study-level factors (ie, quality). These analyses will be conducted when $I^2 < 50\%$ and there are 10 or more studies (n>10) to evaluate differences by specific a priori subgroups.

All analyses will be conducted in RStudio V.2023.06.0+421 for macOS Sonoma 14.1.2 (meta, metasen, metafor packages).

**Contributors** AT is the guarantor. AT, GMH and BRO conceived the initial idea for the systematic review and design of the protocol. AT drafted the manuscript, and GMH and BRO provided critical insight. AT, GH, BRO, EGL and BW contributed to the revision of the manuscript and approved the final version.

**Funding** This work is funded by the UK Research and Innovation, Economic and Social Research Council (grant number: ES/P000703/1).

**Competing interests** None declared.

**Patient and public involvement** Patients and/or the public were not involved in the design, conduct, reporting or dissemination plans of this research.

**Patient consent for publication** Not applicable.

**Provenance and peer review** Not commissioned; externally peer reviewed.

**ORCID iD**
Abigail Thomson http://orcid.org/0000-0001-5322-8488

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
