## [Reviewer comments · BMJ Open]

ARTICLE DETAILS

TITLE (PROVISIONAL)	Self-Directed Digital Interventions for the Improvement of Emotion Regulation - Effectiveness for Mental Health and Functioning in Adolescents: Protocol for a Systematic Review
AUTHORS	Thomson, Abigail; Lawrence, Erin; Oliver, Bonamy; Wright, Ben; Hosang, Georgina M.

VERSION 1 – REVIEW

REVIEWER	Woodcock, Kate University of Birmingham, School of Psychology I have outlined by connection with this area explicitly in the review. I do not believe this constitutes a competing interest but my review will inevitably have been influenced by my experience in the area.
REVIEW RETURNED	01-Dec-2023

GENERAL COMMENTS	The manuscript describes a protocol for a systematic review of self-directed digital interventions for adolescents, which address emotion regulation. The protocol makes a strong case for the review and follows appropriate guidelines for review conduct. However, some important issues arise. My group has recently published a systematic review and meta-analysis on digital interventions for emotion regulation in young people (Reynard, Mitic, Schrank & Woodcock, 2022). Whilst the review proposed poses research questions which are not identical to this previous review, there is nevertheless substantial overlap. It therefore seems critical to frame the present proposal with respect with what it will contribute to the literature over and above this already published review. A long list of databases are listed for inclusion in the search strategy. However, these have a strong medical focus. Given the digital focus of the review, it seems important to justify why no technology-focused databases are included, which often overlap to a limited degree with the more medical oriented databases. For example ACM Digital Library, and IEEE Xplore. Part of the inclusion criteria is that the intervention be "self-directed". Additional exploration and definition of this criterion seems important. For example, would an app for emotion regulation practice which has an adolescent facing part and a caregiver facing part, and includes a focus on effective co-regulation of emotion, be eligible? Our experience in this area is consistent with the authors' expectation that the literature will be highly heterogenous. I think it is important to describe the specific plans with respect to meta-analysis. What threshold would authors require to include studies in a meta-analysis? Under which circumstances would sub-analyses
---

	be conducted? How will the results of the quality appraisal of the included studies be integrated into the planned analyses? Reynard S, Dias J, Mitic M, Schrank B, Woodcock KA. Digital interventions for emotion regulation in children and early adolescents: systematic review and meta-analysis. JMIR Serious Games. 2022 Aug 19;10(3):e31456.
--	---

REVIEWER	Wanjari, Mayur B. Datta Meghe Institute of Medical Sciences, Research and Development
REVIEW RETURNED	28-Dec-2023

GENERAL COMMENTS	Good research questions and protocol of systematic review help to research the body of the current knowledge. Make minor grammatical corrections in the manuscript and add the statistical software name to the methodology section's version.
--

REVIEWER	Bengesai, Annah V University of KwaZulu-Natal
REVIEW RETURNED	16-Jan-2024

GENERAL COMMENTS	Thank you for the opportunity to review this protocol. I found it to be well written with a sound methodology well-articulated. My comments mainly relate to the justification for the review. The introduction does not provide enough justification for conducting a systematic review. A systematic review might not be the most appropriate method if there is a lack of existing knowledge on the topic. The authors should consider revising this to clarify the need for the review. I therefore recommend that the authors add a paragraph highlighting the need for the review. What gaps will it address, and how a systematic review will contribute to advancing knowledge? The methodology for the review is well articulated. The search and inclusion criteria are clear and align with the PRISMA reporting guidelines. I believe the study is important – and strengthening the justification will underscore its importance
--

VERSION 1 – AUTHOR RESPONSE

Comments from Reviewer One

Comment 1: My group has recently published a systematic review and meta-analysis on digital interventions for emotion regulation in young people (Reynard, Mitic, Schrank & Woodcock, 2022). Whilst the review proposed poses research questions which are not identical to this previous review, there is nevertheless substantial overlap. It therefore seems critical to frame the present proposal with respect to what it will contribute to the literature over and above this already published review.

Response: Thank you for highlighting this important systematic review – I agree that there is some overlap in the proposed focus. Still, I feel confident that our manuscript and planned review provide a

novel contribution to the field over and above what has been previously published. Primarily, our review takes a more specific focus to compare the effectiveness of only those self-directed digital interventions targeting emotion regulation (e.g. mobile apps). We feel that synthesising evidence on this specific intervention format is vital, especially given the increasing number of self-directed digital interventions becoming available to adolescents globally, coupled with the lack of current knowledge on the many different components and their effectiveness in supporting young people with emotion regulation.

That being said, I feel that the review published by your group is a valuable contribution and provides substantial evidence for this study to build upon. Therefore, we have made sure to include a sentence describing this contribution in our introduction in an attempt to frame our review more clearly in light of these findings and other previously published reviews with a similar focus (page 6, lines 11-13; page 7, lines 13-14)

Comment 2: A long list of databases are listed for inclusion in the search strategy. However, these have a strong medical focus. Given the digital focus of the review, it seems important to justify why no technology-focused databases are included, which often overlap to a limited degree with the more medical oriented databases. For example ACM Digital Library, and IEEE Xplore.

Response: Thank you for raising this important point. We agree that there is a need to address the strong medical focus of these databases. The databases selected for this review were chosen in collaboration with an information scientist and were based on previous systematic reviews on this subject (interventions for emotion regulation). In developing the search strategy, we aimed to be as comprehensive as possible to ensure we access all suitable papers and current interventions. However, I would agree that not including technology-focused databases may limit our search. Given this is the protocol for our review, and we are still planning to carry out searches before the findings are finalised, we have decided to include ACM Digital Library in our search to ensure that we do not miss any important studies published outside of the medical remit. IEEE Xplore has not been included, as several preliminary searches of this database found 0 results. We have accordingly updated the list of included databases in our methodology (page 10, lines 3-6). We have also updated our Supplementary Materials (search strategy) to include the planned search strategy for this new database.

Comment 3: Part of the inclusion criteria is that the intervention be "self-directed". Additional exploration and definition of this criterion seems important. For example, would an app for emotion regulation practice which has an adolescent facing part and a caregiver facing part, and includes a focus on effective co-regulation of emotion, be eligible?

Response: Thank you – we agree that this criterion needs further clarification. This “self-directed” format, as it is understood in this review, has now been fully defined in the eligibility criteria in Table 1 (Inclusion and Exclusion Criteria). We also felt it was important to include a specific definition of self-directed interventions in the introduction, as this was missing previously (page 6, lines 20-22). We agree that there is some nuance around the self-directed nature of digital interventions and have endeavoured to make this construct clearer throughout the manuscript.

Comment 4: Our experience in this area is consistent with the authors' expectation that the literature will be highly heterogenous. I think it is important to describe the specific plans with respect to meta-analysis. What threshold would authors require to include studies in a meta-analysis? Under which circumstances would sub-analyses be conducted?

Response: Many thanks for highlighting this missing information. We have added more detail to our methodology to describe the specific plans with respect to our planned meta-analyses (page 12, lines

25-28; page 13, line 1). This includes details on the thresholds required to include studies in a meta-analysis (i.e. study quality) or run a meta-analysis at all (i.e. heterogeneity threshold). We also outline the circumstances under which planned sub-group analyses will be conducted (page 13, lines 6-11).

Comment 5: How will the results of the quality appraisal of the included studies be integrated into the planned analyses?

Response: We have clarified this point in the manuscript (page 12, line 28; page 13, line 1). Overall, we plan to exclude any studies judged to be of low quality from the planned meta-analysis. If a meta-analysis is not possible, a narrative synthesis will be employed instead, and the quality of included studies will be reported to inform judgements about the strength of the findings.

Comments from Reviewer Two

Comment 1: Make minor grammatical corrections in the manuscript and add the statistical software name to the methodology section's version.

Response: Thank you for this feedback. Any grammatical errors have been corrected. The statistical software name and programs have also been added to the methodology section (page 13, lines 13-14).

Comments from Reviewer Three

Comment 1: The introduction does not provide enough justification for conducting a systematic review. A systematic review might not be the most appropriate method if there is a lack of existing knowledge on the topic. The authors should consider revising this to clarify the need for the review. I therefore recommend that the authors add a paragraph highlighting the need for the review. What gaps will it address, and how a systematic review will contribute to advancing knowledge?

Response: Thank you very much for raising this point. We agree that further clarity is needed to justify the chosen methodology and the importance of this systematic review. A systematic review was deemed the most appropriate method for synthesis, over and above a scoping or literature review, due to its ability to synthesise a body of evidence to address the specific research questions in this study – i.e. the effectiveness of self-directed digital interventions for emotion regulation. Additional details have been added to clarify this and to illustrate the need for the review and the gaps it will address (page 7, lines 5-25). We hope that these additions demonstrate what existing evidence this systematic review builds upon and how our findings will increase knowledge within the field.

Additional clarifications

In addition to the above comments, a full spelling and grammar check has been carried out, and any errors have been corrected. I look forward to hearing from you in due time regarding my submission. Thank you for your careful consideration of this manuscript.

VERSION 2 – REVIEW

REVIEWER	Woodcock, Kate University of Birmingham, School of Psychology
REVIEW RETURNED	29-Feb-2024

GENERAL COMMENTS	The authors have comprehensively addressed the issues I raised previously. Looking forward to reading the final published manuscript when it is ready.
---